# Horizontal seed dispersal by dung beetles reduced seed and seedling clumping, but did not increase short-term seedling establishment

**Lina Adonay Urrea-Galeano**[1,2]*, **Ellen Andresen**[1]*, **Rosamond Coates**[3], **Francisco Mora Ardila**[1], **Alfonso Díaz Rojas**[4], **Gabriel Ramos-Fernández**[5]

1 Instituto de Investigaciones en Ecosistemas y Sustentabilidad, Universidad Nacional Autónoma de México, Morelia, Michoacán, Mexico, 2 Posgrado en Ciencias Biológicas, Universidad Nacional Autónoma de México, Ciudad de México, Mexico, 3 Estación de Biología Tropical Los Tuxtlas, Instituto de Biología, Universidad Nacional Autónoma de México, Veracruz, Mexico, 4 Instituto de Ecología, A.C., Xalapa, Veracruz, Mexico, 5 Centro de Ciencias de la Complejidad, Universidad Nacional Autónoma de México, Ciudad de México, Mexico

* andresen@cieco.unam.mx (EA); linadonay@hotmail.com (LAUG)

**Data Availability Statement:** All relevant data are within the manuscript and its Supporting Information files.

## Abstract

Dung beetles are secondary seed dispersers, incidentally moving many of the seeds defecated by mammals vertically (seed burial) and/or horizontally as they process and relocate dung. Although several studies have quantified this ecological function of dung beetles, very few have followed seed fate until seedling establishment, and most of these have focused on the effects of seed burial. We know very little about the effects of horizontal seed movement by dung beetles, though it is generally assumed that it will affect plant recruitment positively through diminishing seed clumping. The objective of our study was to assess the effects of dung beetle activity on the spatial distribution of seeds and seedlings, and on the probability of seedling establishment. In a tropical rainforest in Mexico we carried out two complementary field experiments for each of two tree species (*Bursera simaruba* and *Poulsenia armata*), using seeds experimentally imbedded in pig dung and recording their fate and spatial location over time. For both species, dung beetle activity reduced the spatial clumping of seeds and seedlings; however, it did not increase the probability of seedling establishment. We discuss the context- and species-specificity of the combined effects of horizontal and vertical dispersal of seeds by dung beetles, and the need to quantify long-term seedling fates to more accurately determine the effects of seed movement by dung beetles on plant recruitment.

## Introduction

Any movement of a seed after deposition by the primary seed dispersal vector constitutes secondary seed dispersal, a common process that affects seed fate and ultimately plant demography [1,2]. In diplochorous systems, where the primary and secondary dispersal vectors are

**Funding:** EA received a research grant (IN207816) from the Programa de Apoyo a Proyectos de Investigación e Innovación Tecnológica at the Universidad Nacional Autónoma de México (PAPIIT–UNAM; http://dgapa.unam.mx/index.php/impulso-a-la-investigacion/papiit). LAUG received a graduate-study fellowship (294513) from Mexico's Consejo Nacional de Ciencia y Tecnología (CONACyT; https://www.conacyt.gob.mx/). The funders had no role in study design, data collection and analysis, decision to publish, or preparation of the manuscript.

**Competing interests:** The authors have declared that no competing interests exist.

different, a potential advantage of the second phase of dispersal can be the movement of the seeds to microsites that are predictably more adequate for seed survival and/or seedling establishment and survival (i.e., directed dispersal; [3,4]). In tropical forests, a very large proportion of plant species are dispersed primarily by frugivorous animals, mostly birds and mammals [5,6]. In the case of seeds dispersed through mammal defecation, a common diplochorous system occurs, in which dung beetles are often responsible for the secondary dispersal of those seeds [4,7,8].

Dung beetles (Coleoptera: Scarabaeidae: Scarabaeinae) disperse seeds accidentally, as they process dung for feeding or ovipositing [9]. The way in which dung beetles process dung depends on the functional group they belong to. They are classified as dwellers, when all their activities are done in the dung pad or immediately underneath it; tunnelers, if they first construct tunnels in the soil under or near the dung pad and then push and pull portions of dung into the tunnels; and rollers, when they first shape dung portions into balls, which they then roll away from the dung pad to a location where they build a tunnel to bury the dung balls [9]. Competition for the ephimerous and patchily-distributed dung is intense, and in tropical rainforests all the fecal material is quickly relocated beneath the soil by tunneler and roller beetles [10]. Dung relocation is responsible for many of the ecological functions attributed to dung beetles [11], including the secondary dispersal of seeds embedded in the dung [7].

The interaction between dung beetles and defecated seeds can have strong effects on seed fate and seedling establishment, but the direction (positive vs. negative) and intensity of these effects are highly species- and context-specific [8,12,13]. For example, burial by beetles is known to greatly diminish the probability of seed predation [7,14–16]. However, while seed burial by beetles may increase seedling establishment for some plant species due to enhanced seed survival [8,12,17,18], others may be affected negatively due to reduced seedling emergence when seeds are buried too deeply [12,13,19]. Even though secondary seed dispersal by dung beetles has been relatively well studied in tropical forests, our ability to generalize on the ultimate net effect of the seed-beetle interaction on plant fitness is still limited, not only due to the context-dependence of the interaction's result, but also due to the fact that very few studies have followed seed fate until seedling establishment [8,12,13,18,20].

Secondary seed dispersal by dung beetles can include both seed burial (vertical dispersal) and/or horizontal seed dispersal [21], but most studies have focused on the effects of the former. However, it has often been suggested that horizontal movement of seeds by dung beetles, although generally restricted to small distances (< 5 m), may favor plant recruitment by decreasing the degree of spatial clumping in which defecated seeds are deposited by the primary disperser [14,21,22]. Spatial clumping of seeds and seedlings, generally associated with high densities, is known to negatively affect plants through various mechanisms, including decreased germination and increased competition, pathogen attack, predation and herbivory [2,23,24]. To our knowledge, only one study has so far quantified a decrease in the spatial clumping of seedlings of two species, following dung beetle activity, with an associated positive effect on the probability of seedling establishment for one of the species, but not for the other one [20]. However, this study used mixtures of both seed species in each dung pile, and seedling establishment could have been affected by interspecific competition [20]. Thus, considering that: (i) the outcome of the seed-beetle interaction is species- and context-specific, (ii) very few studies have followed seed fate until seedling establishment, and (iii) only one study has quantified the effects of horizontal seed dispersal by dung beetles on seedlings, we believe that additional information is necessary before we can reach a more general conclusion on when secondary dispersal by dung beetles has positive effects on plant regeneration, and when not.

Our main objective was to assess, for two tree species, the effects of secondary seed dispersal by dung beetles, with emphasis on the horizontal movement of seeds, on the spatial distribution of seeds and seedlings, and on the probability of seedling establishment. Our hypothesis

was that dung beetle activity would favor seedling establishment because it reduces seed and seedling clustering through horizontal relocation of the seeds present in dung. To test this hypothesis we carried out field experiments to assess the following predictions: (i) dung beetle activity increases the horizontal distance between adjacent seeds deposited in dung, regardless of seed burial; (ii) dung beetle activity increases the distance between established seedlings; (iii) dung beetle activity increases the probability of seedling establishment.

## Methods

### Study site

The study was carried out in the Los Tuxtlas Biological Station (LTBS), in the Mexican state of Veracruz (18°35′5″ N, 95°4′34″ W; *ca.* 150 m a.s.l.). The LTBS, a protected area established in 1967, encompasses 640 ha of tropical rainforest that constitute, since 1998, one of the core areas of the 155,122-ha Los Tuxtlas Biosphere Reserve [25]. All field activities were conducted under full permission of the authorities of the LTBS, Institute of Biology, National Autonomous University of Mexico. No endangered or protected species were collected. Mean annual temperature is 24.1˚C and mean annual rainfall is 4201 mm, with a short drier period (< 100 mm per month) from March to April, and a long wetter period (≥ 100 mm per month) from May through February, the latter representing *ca.* 95% of the total annual rainfall [26].

Despite intense forest loss and fragmentation occurring in the Los Tuxtlas region, the LTBS and the other core areas of the Biosphere Reserve are an important refuge for animals, including dung beetles, with at least 34 species reported for LTBS [27], as well as many mammals [28,29]. Among mammals present at LTBS that are primary dispersers of seeds through defecation, figure most prominently the herbivorous-frugivorous howler monkeys (*Alouatta palliata*; [14]), but also other important omnivorous-frugivorous species such as coatis (*Nasua narica*), raccoons (*Procyon lotor*), tayras (*Eira barbara*), peccaries (*Pecari tajacu*) and kinkajous (*Potos flavus*) [30].

### Effects of dung beetle activity on seed and seedling spatial distribution and on seedling establishment

To assess the effects of dung beetle activity on seeds and seedlings we carried out two complementary field experiments focusing on two tree species that are abundant in our study site (G. Ibarra-Manríquez, pers. comm.): *Bursera simaruba* (L.) Sarg. (Burseraceae) and *Poulsenia armata* (Miq.) Standl. (Moraceae), hereafter referred to by their genus names. Both experiments were carried out independently for each species.

*Bursera* is a dioecious canopy tree; it is a light-demanding species that grows mostly in treefall gaps, reaching heights of 30 m [31]. Fruiting occurs between October and May; fruits are dehiscent drupes that contain a single seed covered by a juicy aril [31,32]. *Bursera* seeds are 7–9 mm long, 5–7 mm thick and wide, and germinate in ~ 2 weeks [33]. *Poulsenia* is a monoecious late-successional tree species that can reach heights of 20–40 m [34,35]. Fruiting occurs between May and November; the fruit is soft and berry-like, containing 9–17 seeds [32]. *Poulsenia* seeds are 8–10 mm long, 5–7 mm thick and wide, and germinate in ~ 4 weeks [33].

We chose to focus on *Bursera* and *Poulsenia* because they can be dispersed through defecation by mammals (e.g., *Alouatta palliata*; [14]), because their seeds have no dormancy [36], and due to the ease of obtaining the necessary number of seeds for the experiments. Furthermore, the seeds of these two species are of a size large enough to allow marking (see below, Experiment 1), but small enough that beetles at our study site will secondarily disperse many of them while processing dung.

First, we carried out Experiment 2 and then Experiment 1; however, since the former corresponds to the seedling stage and the latter to the seed stage, we describe them in life-cycle order. All experimental seeds of *Bursera* and *Poulsenia* were obtained from freshly fallen fruits underneath parent trees in April and June 2017, respectively. For both species, half of the seeds were used within 2 days of collection to set up Experiment 2; the rest of the seeds were air-dried in the shade and stored at room temperature until the setup of Experiment 1 (six months of storage for *Bursera* seeds and four months for *Poulsenia* seeds). We did not test the viability of the seeds prior to our experiments, and seed storage might have negatively affected it. However, in Experiment 1 we did not require seeds to be viable as we only recorded seed condition and position after 48 h (see below).

In the experiments we used fresh domestic pig dung (collected from a nearby household on the same day it was used) to make experimental dung piles containing seeds, thus mimicking primary seed dispersal through mammal defecation. We used domestic pig dung because we needed large quantities of dung for our experiments, and because it is efficient in attracting rainforest dung beetles [37].

**Experiment 1. Seeds: Secondary dispersal and spatial distribution.** We established 30 experimental sites in the forest understory ($\geq$ 50 m apart and $\geq$ 10 m from fruiting adults of the focal plant species). In each site we had 3 circular plots 50 cm in diameter, with 2–3 m between adjacent plots. The border of each plot was delimited by burying a 30 cm wide metallic mosquito netting strip 10 cm into the soil (S1 Fig). From the inside of each plot we removed the few existing seedlings. However, we kept the leaf litter to avoid affecting the behavior of roller dung beetles, which often choose a spot hidden under litter to build their tunnel (E. Andresen, pers. obs.), and thus litter removal may cause them to roll the dung balls larger distances. Each plot was randomly assigned to one of the following treatment levels: (1) 50 g of dung with 20 seeds embedded in it and access to dung beetles (+Feces+Beetles), (2) 50 g of dung with 20 seeds but no access to dung beetles (+Feces−Beetles), (3) no dung, 20 seeds placed directly on the forest floor (−Feces−Beetles). Since the amount of seeds present in the defecations of rainforest mammals can vary tremendously, depending on the plant and animal species (e.g., [8,38,39]), we used seed numbers that can commonly be found in howler-monkey dung piles (e.g., [40,41]). Furthermore, to better mimic the generally smaller size of individual dung piles of frugivorous rainforest mammals that fall on the forest floor (5–25 g; [22,41,42]) we divided the 50 g of dung into 4 equal portions (S1A Fig), each containing 5 seeds. Portions were placed in the center of plots, with ~ 3 cm between portions; this layout was also used for seeds without dung (S1B Fig). In order to measure seed movement, we marked each seed by threading a 30 cm-long piece of fishing line through it (S1C and S1D Fig). After placing dung and seeds in the plots we covered the two control plots (+Feces−Beetles and −Feces−Beetles) with mosquito netting to exclude dung beetles (S1E Fig), but left the plots with dung beetle access uncovered (+Feces+Beetles). After 48 h, when all dung in the plots with dung beetle activity had disappeared from the soil surface (S1F Fig), we used a 2 cm grid to map the location of each seed in the three plots (S1G Fig); for seeds that were buried we mapped its location projected to the surface. To describe the short-term fate of the seeds we classified each seed into one of three categories: (1) seed intact, when seeds were unharmed; (2) seed predated, when seed remains were found; and (3) seed removed, when the seed could not be found. For seeds intact we also recorded if the seed was visible on the soil surface, under leaf-litter, or buried. In the case of buried and horizontally moved seeds we measured the vertical and horizontal distances to the nearest centimeter. Dung beetle movement was limited by the plot's fence, i.e., seeds could not be dispersed beyond the fence. While this allowed us to find most seeds, it probably caused some underestimation of horizontal distances (see Discussion). This experiment was carried out during the rainy season (October 2017), first with *Poulsenia* seeds, and

after two weeks, the same sites and plots were used to repeat the experiment with *Bursera* seeds.

**Experiment 2. Seedlings: Spatial distribution and probability of establishment.** We used a similar experimental setup as above, but without thread-marking the experimental seeds, to allow for germination. We used the same 30 sites as in Experiment 1, but with different plots for each plant species. For *Bursera*, the experiment started in April 2017 (one of the driest months) and for *Poulsenia* in June 2017 (the beginning of the rainy season). Control plots were immediately covered with mosquito netting, while the plot with dung beetle activity was kept open. After 48 hours the latter plots were also covered, to have the same conditions affecting seed and seedling fates in all treatments. The netting remained throughout the experiment to avoid seed rain and to minimize seed removal by granivorous animals and seedling loss due to herbivory. Once seedling establishment occurred (20 and 32 days after the setup of the experiments for *Bursera* and *Poulsenia*, respectively; S1H and S1I Fig) we checked each plot once a week for 15 weeks. During each check we registered established seedlings of the focal species and we mapped the location of each seedling using the 2 cm grid. We assumed that all seedlings of the focal plant species that we recorded, originated from our experimental seeds because: (i) all plots were > 10 m away from any fruiting adult, and (ii) in a previous study in the same sites and with the same treatments, only two seedlings of *Bursera* and two of *Poulsenia* established, overall, from the soil seed bank during a time period of 8 months [33]. Finally, we acknowledge that using seeds extracted from fruits may yield different results compared to using seeds that have passed through the digestive system of a mammal. However, we expect that whatever difference there might be in terms of seed germination would equally have affected our three treatment levels.

## Data analyses

To measure the spatial distribution of seeds (Experiment 1) and seedlings (Experiment 2), following Lawson *et al.* [20] we used the Clark-Evans nearest neighbor index *R* [43]: $R = \bar{r}_A / \bar{r}_E$, where $\bar{r}_A$ is the average observed distance from an individual to its nearest neighbor in the plot, and $\bar{r}_E$ is the expected mean distance between neighbors if the distribution were random. When *R* < 1 the spatial distribution of individuals is clumped, when *R* = 1 it is random, and when *R* > 1 it is overdispersed [43].

For Experiment 1 we lost one *Poulsenia* plot with the +Feces+Beetles treatment. The nearest neighbor index data were analyzed by fitting linear mixed models (LMMs), one for each species. Treatment (+Feces+Beetles, +Feces−Beetles, and −Feces−Beetles) was the fixed factor, while site was included as a random factor.

For Experiment 2, following Lawson *et al.* [20] we analyzed the values of the nearest neighbor index observed during the week of peak seedling abundance, which was determined separately for each species-treatment combination. We chose this approach because for *Bursera* the temporal pattern of seedling emergence differed strongly among treatment levels, and mortality of seedlings occurred rapidly after emergence (S2 Fig). The weeks of peak abundance were as follows: *Bursera*, +Feces+Beetles: 4 wk, +Feces−Beetles: 9 wk, −Feces−Beetles: 10 wk; *Poulsenia*, +Feces+Beetles: 11 wk, +Feces−Beetles and −Feces−Beetles: 6 wk. The nearest neighbor index data for seedlings was analyzed in the same way as for seeds (LMM) in the case of *Poulsenia*; for *Bursera*, due to singularity problems during model fitting, seemingly caused by almost zero variance estimation for the random effect, we excluded the random factor following Bolker [44] and fitted a simple linear model.

To analyze seedling establishment, we fitted Cox regression models with mixed effects (i.e., frailty models), following [45]. In these models we included treatment (+Feces+Beetles, +Feces

−Beetles, and −Feces−Beetles) as the fixed factor, and site and plot as random factors. For these survival analyses we used as response variable the number of days elapsed until seedling establishment occurred for each seed; when this event did not occur at the end of the experiment, we considered this observation as a right-censored datum (e.g., [46]).

Data analyses were carried out using the R statistical environment (v. 3.5.2; [47]). The nearest neighbor indices were calculated using the function *clarkevans.test* of package 'spatstat' [48]. Models for the nearest neighbor index were fitted using functions *lm* of package 'stats' [47], and *lmer* of package 'lme4' [49]. Models for seedling establishment were fitted with function *coxme* of package 'coxme' [50]. In all models, treatment effect was tested through a Wald Chi-square test using the *Anova* function in the 'car' package [51]. We obtained marginal mean and standard error values, and carried out *post hoc* tests using the function *emmeans* of the package 'emmeans' [52]. We adjusted *P*-values in all *post hoc* tests using the False Discovery Rate method [53], because this method controls for false positives while also minimizing false negatives (e.g., [54]). All processed and raw data used in this study are provided in Supporting Information.

## Results

### Experiment 1. Seeds: Secondary dispersal and spatial distribution

The percentages of experimental seeds lost due to disappearance and predation were very low (*Bursera*: 0.5% and 0.2%, respectively; *Poulsenia*: 2.1% and 0.2%, respectively). For seeds classified as 'intact', most were seeds buried by beetles (*Bursera*: 57.2% ± 26.7%; *Poulsenia*: 54.1% ± 26.4%; mean ± SD), followed by seeds visible on the soil surface (*Bursera*: 32.4% ± 25%; *Poulsenia*: 34.1% ± 25.3%), and by seeds hidden under the leaf litter (*Bursera*: 10.4% ± 11.5%; *Poulsenia*: 11.8% ± 12.4%). For buried seeds, mean depth for both species was 5 cm, but the distribution was asymmetrical with most seeds buried shallowly, although some were buried deeply (*Bursera*, N = 341, min = 1 cm, max = 30 cm, median = 3 cm; *Poulsenia*, N = 308, min = 1 cm, max = 25 cm, median = 2 cm, S3 Fig).

Of all seeds classified as 'intact', 97% of *Bursera* seeds and 98% of *Poulsenia* seeds were moved horizontally at least 2 cm by dung beetles. The mean dispersal distance was 6 cm for both species, again with an asymmetrical distribution (*Bursera*, N = 577, min = 2 cm, max = 24 cm, median = 5 cm; *Poulsenia*, N = 556, min = 2 cm, max = 25 cm, median = 5 cm, S3 Fig).

As we predicted, for both species the mean nearest distance between two seeds was higher in plots with dung beetle activity (*Bursera*: 1.23 cm ± 0.61 cm; *Poulsenia*: 1.15 cm ± 0.51 cm), compared to plots with dung added but beetles excluded (*Bursera*: 0.003 cm ± 0.02 cm; *Poulsenia*: 0.0 cm) and to plots with no dung or beetles (*Bursera*: 0.13 cm ± 0.57 cm; *Poulsenia*: 0.0 cm). The treatment had a significant effect on the spatial distribution of both seed species (*Bursera*, $\chi2$ = 137.46, df = 2, *P* < 0.001, Fig 1A; *Poulsenia*, $\chi2$ = 310.66, df = 2, *P* < 0.001, Fig 1B). Although in all cases the nearest neighbor index values were <1, indicating spatial clustering of seeds, dung beetle activity significantly reduced the degree of seed aggregation, compared to both treatment levels with no dung beetles (*Bursera*, +Feces+Beetles vs. +Feces−Beetles: *t* = 10.60, df = 87, *P* < 0.001, +Feces+Beetles vs.−Feces−Beetles: *t* = 9.64, df = 87, *P* < 0.001; *Poulsenia*, +Feces+Beetles vs. +Feces−Beetles: *t* = 15.30, df = 86, *P* < 0.001, +Feces+Beetles vs.−Feces−Beetles: *t* = 15.30, df = 86, *P* < 0.001). The two control treatment levels were not significantly different from each other (*Bursera*, +Feces−Beetles vs.−Feces−Beetles: *t* = -0.95, df = 87, *P* = 0.34; *Poulsenia*, +Feces−Beetles vs.−Feces−Beetles: *t* = 0, df = 86, *P* = 1).

### Experiment 2. Seedlings: Spatial distribution and probability of establishment

As observed for seeds, for both plant species we found that the nearest neighbor distance between seedlings was higher in plots with dung beetle activity (*Bursera*, 3.67 cm ± 1.43 cm;

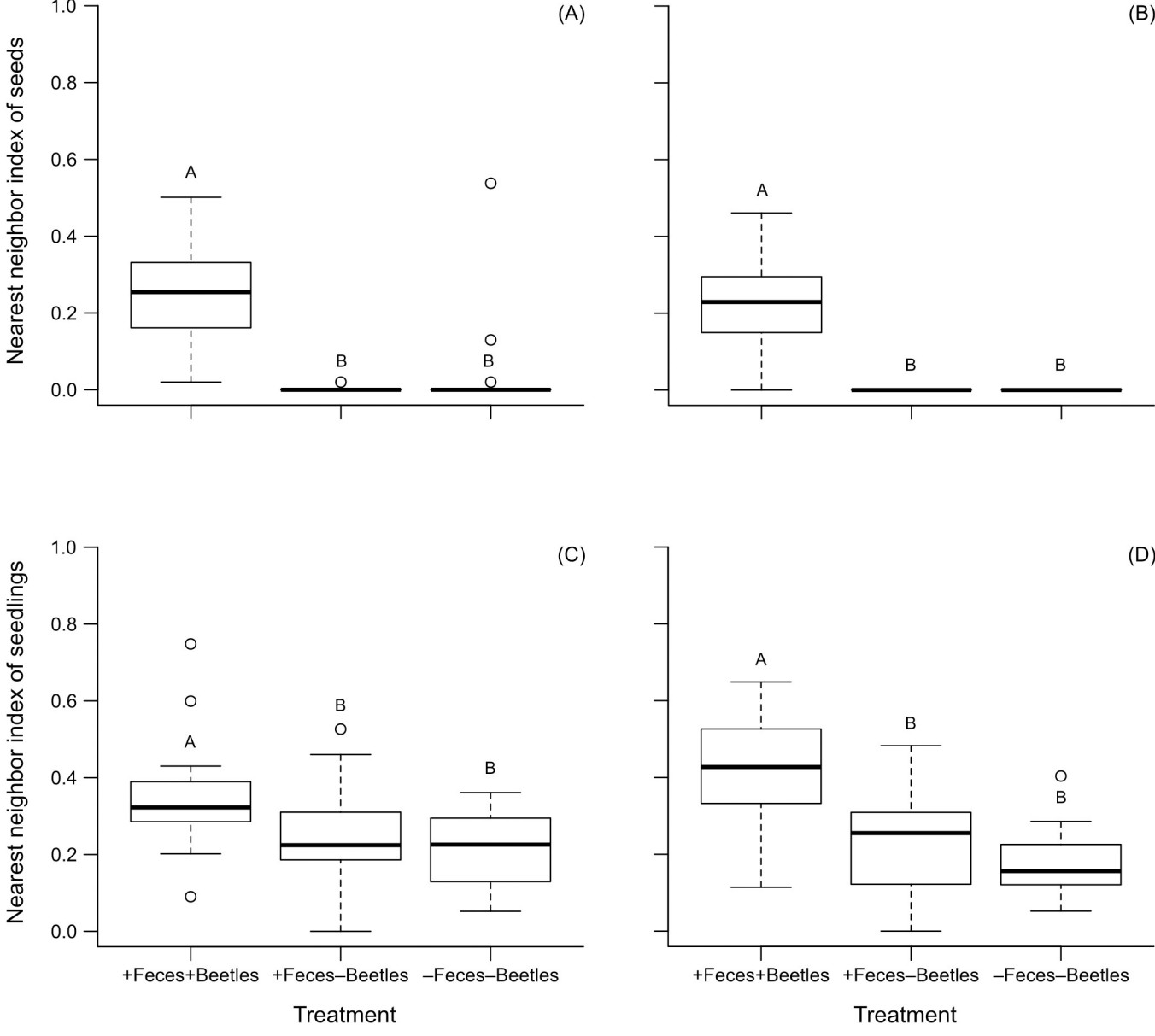

**Fig 1. Box-plots of the nearest neighbor index for seeds and seedlings of two plant species.** The index was measured for *Bursera* seeds (A) and seedlings (C) and *Poulsenia* seeds (B) and seedlings (D) in plots (50-cm-diameter) with three treatment levels: 50 g of feces and dung beetle access (+Feces+Beetles), 50 g of feces and dung beetle exclusion (+Feces–Beetles), and no feces or dung beetles (–Feces–Beetles). In the first two treatments 20 seeds were mixed in the dung, and in the last treatment seeds were placed on the soil surface. Independent experiments were carried out for seeds and seedlings. Seedling results are from data observed during the week of peak seedling abundance, which was determined separately for each species-treatment level combination (see text for details). Circles represent outliers; different letters above bars indicate statistical differences.

*Poulsenia*, 4.39 cm ± 2.14 cm; S4 Fig), than in plots with dung added but beetles excluded (*Bursera*, 2.79 cm ± 1.84 cm; *Poulsenia*, 2.56 cm ± 1.84 cm) and plots with no dung or beetles (*Bursera*, 2.34 ± 1.54; *Poulsenia*, 1.86 cm ± 1.29 cm). The treatment had a significant effect on the spatial distribution of seedlings of both species (*Bursera*, $F = 5.57$, df = 2, $P = 0.006$, Fig 1C; *Poulsenia*, $\chi2 = 31.88$, df = 2, $P < 0.001$, Fig 1D). As with seeds, dung beetle activity significantly reduced spatial aggregation of seedlings when compared to both control plots (*Bursera*, +Feces+Beetles vs. +Feces–Beetles: $t = 2.26$, df = 57, $P = 0.04$, +Feces+Beetles vs.–Feces–

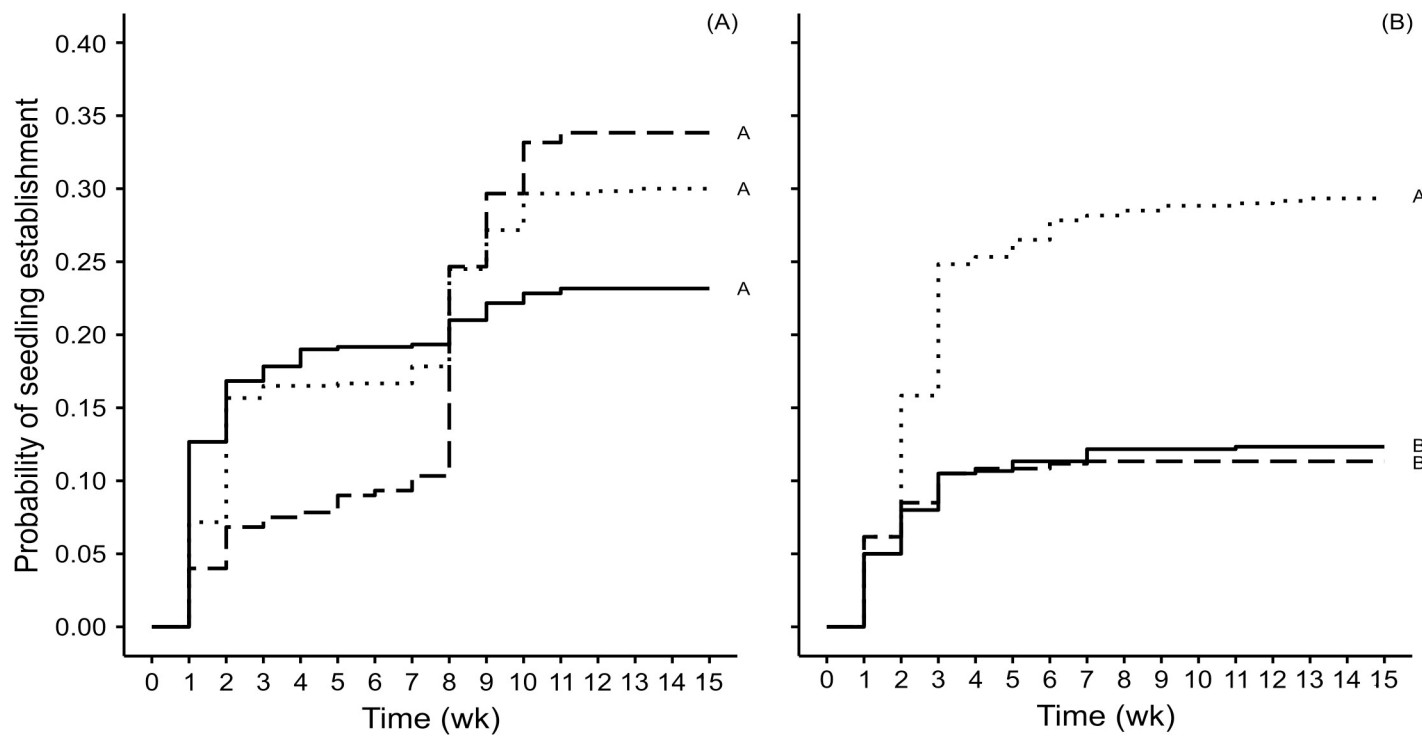

**Fig 2. Kaplan–Meier curves for the probability of seedlings establishing from experimental seeds of two plant species.** During 15 weeks, seedling establishment was monitored for *Bursera* (A) and *Poulsenia* (B) in plots containing: 50 g of feces and dung beetle access (solid black lines), 50 g of feces and dung beetle exclusion (dashed black lines), and without feces or dung beetles (dotted black lines). In the first two treatments 20 seeds were mixed in the dung, and in the last treatment seeds were placed on the soil surface. All curves have censored data. Different letters next to each curve indicate statistical differences.

Beetles: $t$ = 3.30, df = 57, $P$ = 0.005; *Poulsenia*, +Feces+Beetles vs. +Feces–Beetles: $t$ = 3.89, df = 53, $P$ < 0.001, +Feces+Beetles vs.–Feces–Beetles: $t$ = 5.50, df = 53, $P$ < 0.001). The two control treatment levels were not significantly different from each other (*Bursera*, +Feces–Beetles vs.–Feces–Beetles: $t$ = 1.25, df = 57, $P$ = 0.21; *Poulsenia*, +Feces–Beetles vs.–Feces–Beetles: $t$ = 1.10, df = 53, $P$ = 0.27).

Of all experimental seeds, 29% of *Bursera* and 18% of *Poulsenia* seeds registered seedling emergence. During the 15 weeks in which we monitored seedling emergence, *Poulsenia* followed a steady pattern of increase until an asymptote was reached, while for *Bursera* we observed peaks of emergence followed by seedling death (S2 Fig). The treatment did not have a significant effect on the probability of seedling establishment in the case of *Bursera* ($\chi$2 = 4.26, df = 2, $P$ = 0.12; Fig 2A). However, it had a significant effect for *Poulsenia* ($\chi$2 = 69.90, df = 2, $P$ < 0.001); the probability of seedling establishment was higher in the plots with no feces, and it was equally low in the two treatment levels with feces, regardless of dung beetle activity (Fig 2B). In plots with no feces the probability of establishment increased by a factor of 2.70 compared to plots with dung beetle activity.

## Discussion

Almost all experimental seeds (> 97%) were moved by dung beetles horizontally and more than half the seeds (> 55%) were moved vertically (i.e., buried). As predicted, dung beetle activity was associated with a decrease in the spatial clumping of both seeds and seedlings for both plant species. However, contrary to our expectation, dung beetle activity did not have a positive effect on the probability of seedling establishment. While the first result shows clearly

the importance of dung beetle activity in diminishing the aggregation of seeds deposited in fecal clumps by fruit-eating mammals and of the seedlings establishing from those seeds, the second shows that this effect does not necessarily translate into increased seedling establishment. This stresses the need to be cautious when drawing conclusions regarding the effects of dung beetle activity on plant regeneration, based only on their effects on seeds.

Our result on reduced seed aggregation was expected, as several studies have determined and measured horizontal seed movement caused by dung beetles [12,16,21,22,55]. In terms of seedling spatial aggregation, our results are consistent with the only published study (that we are aware of) designed to quantify this consequence of dung beetle activity. In that study, Lawson et al. [20] also found that dung beetle activity decreased seedling spatial clumping in two plant species, *Tabernaemontana donnell-smithii* Rose and *Guazuma ulmifolia* Lam. For seedling establishment, they obtained mixed results: while one of the species had a higher percentage of establishment in plots with dung beetle activity, the other one had lower establishment in those plots [20]. In our study, one species had the highest probability of seedling establishment in plots with dung added but beetles excluded, while the other species had significantly more seedling establishment in plots without dung (Fig 2). Lawson's study used both seed species mixed together in their experiments and the authors argued that their results on seedling establishment might have been affected by competitive interactions between germinating seeds of the two species. Furthermore, they pointed out that given the short duration of their study (4 weeks after experimental setup), they could not disentangle potential mechanisms affecting seedling establishment mediated by dung beetle activity (e.g., effects of dung beetles on seed clumping vs. on seed germination; [20]). We carried out independent experiments for each seed species, and we monitored seedling establishment for a longer time period (15 weeks after onset of establishment), yet we also recognize that several factors could have influenced our results. Both our study and that of Lawson underscore the fact that the outcomes of seed-beetle interactions are species- and stage-specific, and can be affected by many factors [7,8,12,13,33].

So, why did we not find a positive effect of dung beetle activity on short-term seedling establishment? First, vertical seed dispersal by dung beetles might have hindered seedling emergence, counterbalancing a positive effect of decreased clumping due to horizontal dispersal. We found that over half of all experimental seeds in the plots with dung beetle activity were buried. Several studies in tropical forests have shown that, depending on seed species and burial depth, the effect of vertical seed dispersal by dung beetles can promote seedling establishment (through a dominant effect of avoiding seed predation) or it can hinder it (through a dominant effect of non-emergence of germinating seeds; [8,12,13,18,33]). It has been considered that burial depths $\leq$ 3 cm should favor seedling establishment of rainforest seeds dispersed by dung beetles [7]. In our study, the median burial depth was 2–3 cm, meaning that half of the seeds buried by dung beetles were located at greater depths (S3 Fig) and may have suffered from non-emergence or other types of seed mortality associated with burial [56]. In a parallel study in the same study region, in which we buried seeds of the focal plant species at 3, 5 and 10 cm, seedling establishment decreased with depth [33]. Furthermore, in that study, only 7% of *Bursera* and 14% of *Poulsenia* seeds buried at 3 cm emerged as seedlings, compared to 30% for seeds placed on the surface. This clearly indicates that even shallow seed burial depths have a strong negative effect on the recruitment of these two species. However, it could also be that seasonal differences in dung beetle assemblages caused a higher proportion of seeds to be buried and/or buried more deeply in Experiment 2, which was setup in April-June, than in Experiment 1, which was done in October. When we sampled dung beetles, we found higher abundances, particularly of large beetles ($\geq$ 10 cm body length), in April than in October (S1 Table).

Second, it is possible that the decrease in seed and seedling spatial aggregation due to dung beetle activity, though statistically meaningful, might not have the necessary effect size to be of ecological significance. Though most seeds were moved horizontally by dung beetles, dispersal distances were short (~ 80% between 2 and 7 cm; S3 Fig). However, given our experimental setup, these distances could, to a certain degree, be underestimates, as the movement of beetles was limited by the edge of the plot, i.e., the plot radius of 25 cm was the maximum possible distance recorded. In our study site, roller dung beetles have been reported to move dung balls (often containing seeds) to a mean distance of 1.2 m, and up to 5 m [14]. On the other hand, tunneler dung beetles bury the dung relatively close to the dung source [10], but that does not mean that they do not cause any horizontal movement of seeds. Tunnelers sometimes push dung portions on the soil surface for a short distance (< 20 cm) before burying it; furthermore, underground tunnels are often dug with an angle < 90° such that seeds moved into the tunnels are displaced both vertically and horizontally (L.A. Urrea-Galeano and E. Andresen, pers. obs.). Given that 95% of all individuals captured in our pitfall traps were tunnelers (S1 Appendix and S1 Table), we believe that most seed movement we observed was carried out by this functional group and that most horizontal distances recorded were therefore accurate. Nonetheless, the few seeds that can be dispersed longer distances by roller dung beetles may be the ones establishing seedlings that have higher long-term survival probabilities. For example, a recent study found that, while dung beetle activity had a negative effect on seedling emergence of one plant species, it increased seedling survival [13]. Although the authors did not interpret the latter result as a possible consequence of reduced seedling clumping, this explanation remains plausible. Thus, studies following the fate of seeds moved by beetles in an unconstrained fashion and assessing seedling recruitment and survival, will be necessary to better understand the role of horizontal seed movement in decreasing the density-dependent processes causing seed/seedling mortality. Furthermore, since the life-stage at which plants suffer density-dependent mortality varies among species and contexts (e.g., [57]), future research will need to carefully determine the necessary duration of studies, in order to accurately assess the effects of reduced seed/seedling clumping caused by dung beetles, on plant fitness.

Related to the point above, it is important to mention that we protected experimental plots with mosquito netting, thus excluding most seed predators and seedling herbivores, while assuming that a positive effect of decreased clumping on seedling establishment might still be evident due to less intense seed/seedling competition and pathogen attack. Yet, density-dependent seed predation and seedling herbivory are known to be frequent processes affecting plant fitness [2,23,24]. Seed predation can often be very high, with 100% seed loss not being uncommon [58]; we used the netting precisely to avoid losing experimental seeds, and thus be able to have enough remaining seeds and seedlings for data analyses. Therefore, by excluding seed predators we also excluded the known positive effect of seed burial by dung beetles, i.e., seed predation avoidance [14–16,59]. To fully understand the effect of dung beetle activity on seed/seedling fates, we will need to design studies that allow us to simultaneously assess each of the positive and negative effects of horizontal and vertical seed dispersal by beetles, while disentangling the effects of both types of seed movement.

Finally, we want to stress once more how species-specific requirements for seed survival, germination, seedling establishment and survival, can strongly influence the results of the beetle-plant interactions. In our study, though seeds of both focal species were similar in size, a plant attribute that strongly affects the short-term fate of seeds after dung beetle activity (e.g., [33,60]), functional seedling attributes differed between species and were perhaps responsible for the different patterns of seedling establishment. For example, the timing of peak seedling establishment differed between treatment levels in *Bursera* but not in *Poulsenia* (S2 Fig). The temporal dynamics in *Bursera* seedlings was determined by their shade-intolerance [31],

which caused the death of seedlings shortly after emergence. *Poulsenia* seedlings, on the other hand, are shade-tolerant [61], but the seeds suffered clear negative effects, both from remaining imbedded in dung (when beetles were excluded) and from being buried (when beetles were active; Fig 2). In the case of seeds that remained in dung, it is possible that merely dung presence could explain differences between the two species, as a few studies have shown that dung itself can have either positive or negative effects on germination or seed/seedling performance, depending on seed species [62,63]. In the case of seeds moved by dung beetles, many of which were buried, characteristics associated with the position, exposure and function of cotyledons can play an important role in determining the probability of seedlings emerging from buried seeds and surviving through the establishment period [64]. In this regard, *Poulsenia* seedlings are cryptocotylar hypogeal with reserve storage (CHR) while *Bursera* seedlings are phanerocotylar epigeal with foliaceous cotyledons (PEF; [65]). One study found that out of ten rainforest seed species tested, those that were CHR had the highest seedling establishment from buried seeds, while PEF species had the lowest [7]. In a previous study, CHR *Poulsenia* seeds buried at 3–5 cm did indeed have more seedling establishment, compared to PEF *Bursera* seeds [33]. It seems that in the present study, however, the lower seedling establishment observed for *Poulsenia* seeds in plots with dung beetle activity, compared to those of *Bursera*, is due, at least partly, to a cause different than seedling functional morphology. For example, differences between species could be due to seasonality in dung beetle activity, since the experiment for *Bursera* was started in April and the one with *Poulsenia* in June, a drier and a rainier month, respectively. Secondary seed dispersal by dung beetles has been shown to be affected by seasonality [7], not only because dung beetle assemblages vary seasonally [10], but also because in rainier months softer soils might favor deeper seed burial [21], which in turn might hinder seedling emergence.

In conclusion, our study confirms the important role dung beetles may play, through the horizontal secondary dispersal of seeds, in diminishing seed/seedling aggregation after seeds are deposited by mammals in fecal clumps. However, we did not find evidence indicating that this effect may have consequences for early seedling establishment. Longer-term studies will be necessary to ascertain if over time, the decreased seed/seedling clumping translates into increased seedling or sapling survival probabilities. Furthermore, since vertical and horizontal dispersal by dung beetles can simultaneously affect seeds and seedlings through different and sometimes opposing mechanisms, we must design studies that will allow us to assess each of them accurately, but in conjunction. Finally, given the species- and stage-specific outcome of the interactions between plants and dung beetles, more studies, including many seed species and their different stages of regeneration (seed bank, germination, emergence, establishment, survival), are necessary to fully understand the impact of dung beetle activity on plants.

## Supporting information

**S1 Table. Dung beetles captured in the Los Tuxtlas Biological Station, Veracruz, Mexico.** Beetles captured using 10 pitfall traps each baited with 50 g of fresh domestic pig dung and opened during 48 hours. Sampling was conducted in April, September and October 2016. Information about dung relocation behavior (tunneler 'T', roller 'R', dweller 'D'), diet (predominantly feces 'F' or carrion 'C'), and body measurements (dry weight, body length) are from Díaz & Favila [2009. Escarabajos coprófagos y necrófagos (Scarabaeidae, Silphidae y Trogidae) de la reserva de la biosfera Los Tuxtlas, México. Memorias VIII Reunión Latinoamericana de Scarabaeidología (Coleoptera: Scarabaeoidea). Pp. 34. Xalapa, Veracruz]. *Eurysternus* has a unique behavior in which dung is not relocated; unlike typical rollers, feeding takes place directly in the dung source, but unlike typical dwellers, dung balls are made for nesting and

are lightly covered by soil near the dung source [Halffter, G., & Edmonds, W. D. 1982. The nesting behavior of dung beetles (Scarabaeinae). An ecological and evolutive approach (Man and the Biosphere Program, Publication 10). Instituto de Ecología, Mexico City].
(DOCX)

**S1 Fig. Methodological details of the two experiments carried out to assess the effects of dung beetle activity on the spatial distribution of seeds (Experiment 1) and seedlings (Experiment 2), and on the probability of seedling establishment (Experiment 2).** (A) 50 g of fresh domestic pig dung used in plots with dung added; dung was divided into 4 equal portions, each containing 5 seeds of either *Bursera simaruba* or *Poulsenia armata*; (B) seeds inside the plots with no dung added; seeds were placed directly on the soil surface (as indicated by the red arrows); (C and D) experimental seeds of *Poulsenia* thread-marked with a 30 cm-long fishing line in Experiment 1 (for Experiment 2, seeds were not thread-marked); (E) mosquito netting excluded dung beetles from control plots during the first 48 h (both experiments), and also excluded seed rain and seed/seedling predators from all plots after the first 48 h (Experiment 2); (F) plot with dung beetle activity after 48h of having placed the dung piles containing seeds; no dung remains visible on the soil surface; (G) grid (2 cm) used to map the location of each seed (Experiment 1) and seedling (Experiment 2), to calculate the nearest neighbor index; (H and I) seedlings of *Bursera* and *Poulsenia*, respectively, establishing inside plots from experimental seeds.
(DOCX)

**S2 Fig. Mean number of seedlings of two plant species registered weekly for 15 weeks.** *Bursera simaruba* (A) and *Poulsenia armata* (B) seedlings established in 50-cm-diam plots (N = 30 for each species-treatment level) with three treatment levels: 50 g of dung and beetle access (black continuous line), 50 g of dung and beetle exclusion (black dashed line), and with no dung or beetles (black dotted line). In each plot of the first two treatment levels 20 seeds were mixed in the dung, in the last treatment level seeds were placed on the soil surface. Error bars represent ± 1 SE.
(TIF)

**S3 Fig. Frequency distribution of dispersal distances for two seed species.** Vertical (A and B) and horizontal (C and D) dispersal distances for experimental seeds that were secondarily dispersed by dung beetles (seeds that remained in their original position were not dispersed and thus were not included in these graphs). Seed species are *Bursera simaruba* (A and C) and *Poulsenia armata* (B and D). Dung beetle activity was restricted to circular plots 25 cm in radius; inside each plot 50 g of fresh pig dung containing 20 seeds of one species was placed.
(TIF)

**S4 Fig. Mean nearest neighbor distance for seedlings of two plant species over time.** Distance for seedlings of *Bursera simaruba* (A) and *Poulsenia armata* (B) over 15 weeks in plots (N = 30 for each species-treatment level) with three treatment levels: 50 g of dung and beetle access (black continuous line), 50 g of dung and beetle exclusion (black dashed line), and with no dung or beetles (black dotted line). Error bars represent ± 1 SE.
(TIF)

**S1 Appendix. Methods and results of dung beetle sampling in the Los Tuxtlas Biological Station, Veracruz, Mexico.**
(DOCX)

**S1 Dataset. Data on the nearest neighbor index for seeds of *Bursera* in experiment 1.** See Expe1RatioBursera in metadata file.
(XLSX)

**S2 Dataset. Raw data on the coordinates for seeds of *Bursera* in experiment 1.** See Expe1RawdataBursera_Spatialdistribution in metadata file.
(XLSX)

**S3 Dataset. Raw data on the secondary seed dispersal for seeds of *Bursera* in experiment 1.** See Expe1RawdataBursera_Secondaryseeddispersal in metadata file.
(XLSX)

**S4 Dataset. Data on the nearest neighbor index for seeds of *Poulsenia* in experiment 1.** See Expe1RatioPoulsenia in metadata file.
(XLSX)

**S5 Dataset. Raw data on the coordinates for seeds of *Poulsenia* in experiment 1.** See Expe1RawdataPoulsenia_Spatialdistribution in metadata file.
(XLSX)

**S6 Dataset. Raw data on the secondary seed dispersal for seeds of *Poulsenia* in experiment 1.** See Expe1RawdataPoulsenia_Secondaryseeddispersal in metadata file.
(XLSX)

**S7 Dataset. Data on the nearest neighbor index for seedlings of *Bursera* during the week of peak seedling abundance in experiment 2.** See Expe2RatioBursera in metadata file.
(XLSX)

**S8 Dataset. Raw data on the coordinates for seedlings of *Bursera* during the week of peak seedling abundance in experiment 2.** See Expe2RawDataBursera_Spatialdistribution in metadata file.
(XLSX)

**S9 Dataset. Data on the germination of seeds of *Bursera* registered weekly for 15 weeks in experiment 2.** See Expe2SurvivalBursera in metadata file.
(XLSX)

**S10 Dataset. Data on the nearest neighbor index for seedlings of *Poulsenia* during the week of peak seedling abundance in experiment 2.** See Expe2RatioPoulsenia in metadata file.
(XLSX)

**S11 Dataset. Raw data on the coordinates for seedlings of *Poulsenia* during the week of peak seedling abundance in experiment 2.** See Expe2RawdataPoulsenia_Spatialdistribution in metadata file.
(XLSX)

**S12 Dataset. Data on the germination of seeds of *Poulsenia* registered weekly for 15 weeks in experiment 2.** See Expe2SurvivalPoulsenia in metadata file.
(XLSX)

**S13 Dataset. Data on the dung beetles captured in the Los Tuxtlas Biological Station, Veracruz, Mexico.** See S1 Appendix_RawdataDungbeetle_Sampling in metadata file.
(XLSX)

**S1 Metadata. Descriptive information about all datasets.**
(TXT)

## Acknowledgments

We are thankful for the logistical support provided by UNAM's IIES, Posgrado en Ciencias Biológicas (PCBiol), and Estación de Biología Tropical Los Tuxtlas. We thank Sr. Santos Perez Ferman for his help in the field. We thank Ek del Val for providing insightful comments to this research project.

## Author Contributions

**Conceptualization:** Lina Adonay Urrea-Galeano, Ellen Andresen, Gabriel Ramos-Fernández.

**Formal analysis:** Lina Adonay Urrea-Galeano, Francisco Mora Ardila.

**Funding acquisition:** Ellen Andresen.

**Investigation:** Lina Adonay Urrea-Galeano, Alfonso Díaz Rojas.

**Methodology:** Lina Adonay Urrea-Galeano, Ellen Andresen, Rosamond Coates.

**Project administration:** Lina Adonay Urrea-Galeano, Ellen Andresen, Rosamond Coates.

**Resources:** Ellen Andresen, Rosamond Coates, Alfonso Díaz Rojas.

**Supervision:** Lina Adonay Urrea-Galeano, Ellen Andresen, Rosamond Coates, Gabriel Ramos-Fernández.

**Writing – original draft:** Lina Adonay Urrea-Galeano, Ellen Andresen.

**Writing – review & editing:** Lina Adonay Urrea-Galeano, Ellen Andresen, Rosamond Coates, Francisco Mora Ardila, Alfonso Díaz Rojas, Gabriel Ramos-Fernández.

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
