## [Decision Letter · Decision Letter 0]

24 Jun 2019

PONE-D-19-15149

Horizontal seed dispersal by dung beetles reduced seed and seedling clumping, but did not increase short-term seedling establishment

PLOS ONE

Dear Lina Adonay Urrea Galeano,

Thank you for submitting your manuscript to PLOS ONE. After careful consideration, we feel that it has merit but does not fully meet PLOS ONE’s publication criteria as it currently stands. Therefore, we invite you to submit a revised version of the manuscript that addresses the points raised during the review process.

The three reviewers agreed that the manuscript is well presented and is an important contribution to the effect of dung beetle activity on plant recruitment. However, it is necessary that the authors make the comments suggested by the reviewers for further consideration.

We would appreciate receiving your revised manuscript by 22 of september. To enhance the reproducibility of your results, we recommend that if applicable you deposit your laboratory protocols in protocols.io, where a protocol can be assigned its own identifier (DOI) such that it can be cited independently in the future. For instructions see: http://journals.plos.org/plosone/s/submission-guidelines#loc-laboratory-protocols

We look forward to receiving your revised manuscript.

Kind regards,

Guillermo C. Amico

Academic Editor

PLOS ONE

Journal Requirements:

A graduate fellowship was awarded to LAUG by CONACYT (294513).

EA received a research grant from Programa de Apoyo a Proyectos de Investigación e Innovación Tecnológica (PAPIIT–UNAM, project IN207816). The funder had no role in study design, data collection and analysis, decision to publish, or preparation of the manuscript.

Additional Editor Comments:

The three reviewers agreed that the manuscript is well presented and is an important contribution to the effect of dung beetle activity on plant recruitment. However, it is necessary that the authors make the comments suggested by the reviewers for further consideration.

Reviewers' comments:

Reviewer's Responses to Questions

**Comments to the Author**

1. Is the manuscript technically sound, and do the data support the conclusions?

Reviewer #1: Yes

Reviewer #2: Yes

Reviewer #3: Yes

2. Has the statistical analysis been performed appropriately and rigorously? 

Reviewer #1: Yes

Reviewer #2: Yes

Reviewer #3: Yes

3. Have the authors made all data underlying the findings in their manuscript fully available?

Reviewer #1: Yes

Reviewer #2: No

Reviewer #3: Yes

4. Is the manuscript presented in an intelligible fashion and written in standard English?

Reviewer #1: Yes

Reviewer #2: Yes

Reviewer #3: Yes

5. Review Comments to the Author

Reviewer #1: In their manuscript “Horizontal seed dispersal by dung beetles reduced seed and seedling clumping, but did not increase short-term seedling establishment” the authors have tested the effects of horizontal seed movement by dung beetle for two tree species. They found that dung beetle activity reduced the spatial clumping of seeds and seedlings, however, it did not increase the probability of seedling establishment. This study to help fill our knowledge gaps regarding the effects of secondary seed dispersal by dung beetles. The sample design seems adequate and robust to answer their main questions. I also like writing and organization of the manuscripts as well as concise results and discussion. I present below a list of minor comments that will hopefully aid you in improving your manuscript. I therefore really welcome this ms.

L.30- In your abstract and introduction, I think it is necessary that you state more clearly your objectives, rather than “the main goal of study was to help fill this information gap “

L. 31- I recommend adding a sentence here, explained better your field experiments

L. 33 - Be specific: What tree species?

L. 80- I think it is necessary that you state more clearly what the unique contribution of your study is. What is your study diferente from Lawson et al. (2012)?

L. 97- Add elevation of Station

L. 101 - Was the experiment carried out during which season?

L. 107 – 108- References for frugivores are need here, or declare if it is personal observation.

L. 116 – Why did not you use a mixed of human and pig dung?

L. 188 – Were the dung beetles identified using any entomological key?

L. 121 - I would like to know more information about two focal tree species. Are these species dominant in the station? Why do you call these focal species?

L. 106 and L 125: Alouatta palliata or Alouatta palliata mexicana? Try to use the same name.

L.125 - Is Alouatta palliata Mexicana herbivorous or omnivore?

L. 131 - Why as model herbivore? Is the pig dung used as omnivores?

L. 131 - Why you used fresh domestic pig dung rather than of Alouatta palliata? Pigs do not form part of the local biota and are onmnivores (?) rather than frugivores.

L. 137 - I would like to see a photo of experiment in supplemental material.

L. 138 – Why keep the leaf litter?

L.136- Were the plots of all treatments covered?

L. 161- I think that you said the same information in the lines 163 and 164.

L. 165 - Is the dung beetle activity went beyond the plot? Because you said that the plot with dung beetle activity was covered with netting after 48 hours.

L. 215 - What is False Discovery Rate method?

L. 218 - The reason for the dung beetle sampling should be in the methodology

L 219 - 225 – In the results session should have the results regarding their objectives. I suggest you transfer this information on to supplementary material or add a question in your objectives.

L 225 - Why did you put information on Scarabaeidae, Trogidae and Silphidae? In your introduction you state that your study is about dung beetles (scarabaeinae).

L 233 – The sum of Pousenia was 99.9%. Check the values, please.

L. 256, 258, 259 - Put the df

L. 298 - The information “but not in accordance with our prediction” should not be in the results.

L. 316 - In your introduction you describe the tunnelers: they construct tunnels in the

soil beneath the dung pad for burying portions of dung. However, approximately 90% of dung beetle community of your study is tunneler beetles. How do you explain the large amount of seeds moved horizontally (97%)?

L. 325 - I would like you to discuss a bit about what time would be enough. Will this depend on the plant species?

Reviewer #2: Review report PONE-D-19-15149

General comments

The paper covers an interesting topic and certainly fills a gap in knowledge as it is indeed assumed that dung beetles have a positive effect on seed germination and seedling survival. I am not too familiar with the most recent publications in tropical dung beetle ecology, but the paper certainly fills a gap in the current knowledge.

The manuscript is well written in a clear and concise matter and the figures are clear. The used methods are correct, although I do have a suggestion for improving the statistical analysis. I am not sure whether the method is applicable on the dataset that was collected (no raw data were provided), but I’m quite certain that a survival analysis would give more insight in the establishment experiment (experiment 2). Now the authors analyzed the peak emergence data for Bursera, because a large portion of the seedlings died at a given moment. In a survival analysis, the cumulative probability of a certain event (establishment in this case) is calculated. Onofri et al. (2010) wrote a very interesting paper about using survival analysis for the analysis of the germination and emergence of seeds. There is also an R-package available on CRAN with functions for survival analyses (package ‘survival’). By conducting a survival analysis, the authors could assess the differences between treatments over the entire time series instead of using 1 single moment in a long time series.

Data files are provided for all experiments and metadata are provided. However, the raw datasets are not provided, instead derived data (indices, cumulated dung beetle numbers without capture dates) are given. I don’t know the journal’s policy about data availability and the motivation for providing data, but these types of datasets are not very useful. Maybe, the results of this study could become useful in a future study doing a meta-analysis which needs raw data (e.g., real distances instead of the Clark Evans index). If the authors cannot be contacted at that time or they cannot retrieve the raw datasets anymore, sharing the datasets with this paper was not worth the effort.

Also, the dung beetle dataset could provide interesting information for policy makers and conservationalists (e.g., for making red lists), so it would be useful to put it in a suitable database. For biodiversity data, the ‘Global biodiversity information facility (GBIF)’ (www.gbif.org) is the standard database. When publishing data at GBIF, you become the author of the dataset and you get a doi code. So, the dataset can be cited and you get acknowledged for your efforts.

Specific comments

L43: potential advantage to plants of the second phase of… -> remove ‘to plants’ as it is clear that plants are the beneficiary of the process

L61: ‘sign’ sounds a bit weird -> maybe ‘nature’ or ‘(final) result’ ?

L117: Which measures were taken? Which equipment was used and what was the precision of the measurements?

L139: How did you decide to use 20 seeds in dung portions of 50g? How is this related to the seed density in ‘natural’ droppings found in this region?

L152: seed ‘alive’: did you test the viability of the seeds? If not, intact seeds would be a better description.

L159 and following: It is not clear from the text, but did you count the number of Poulsenia and Bursera seedlings originating from the soil seed bank?

L229-232: this part rather belongs to the discussion section.

L232: See my earlier remark. Unless you have tested the retrieved seeds for viability, you should refer to them as intact seeds.

L244-246: this rather belongs to the discussion section

L319-323: It seems that the presence of dung did have a greater effect on the emergence of seedlings than the presence of dung beetles. The fact that dung reduces the germinability of seeds was also found in other studies, e.g., Milotić and Hoffmann (2016). This study was done in a completely different environment and using different plant species (temperate grassland species), but I am not aware of similar studies in a tropical environment.

L338: Another factor to consider is the fact that clean, undigested seeds were used in the experiments. I assume that was practically impossible to use digested seeds as you would have to feed wild animals with seeds and collect the dung afterwards (furthermore, the issue would arise about which species should be used as a ‘digester’ species), but it is worthwhile mentioning this in the discussion. In the natural situation, seeds that passed the gut environment of a frugivore might have an altered germination probability and speed. So, maybe the presence of dung would not be that much of a problem then.

L350 and following: did you test the viability of the seeds prior to the experiments? It could also be that a relatively large proportion of the seeds is dead or dormant. Is there anything known about the germination ecology of these species (e.g., in Baskin and Baskin (2001))?

L393: What is the usual fruiting season for these species or do they carry fruits year-round?

Fig S3: For Bursera, the mean nearest neighbor distance in the +feces+beetles treatment increases over time. Do you have any idea why?

References

Baskin C.C., Baskin J.M. (2001) Seeds: ecology, biogeography and evolution of dormancy and germination. Academic Press, San Diego.

Milotić, T., & Hoffmann, M. (2016). Reduced germination success of temperate grassland seeds sown in dung: consequences for post‐dispersal seed fate. Plant Biology, 18(6), 1038-1047.

Onofri A., Gresta F., Tei F. (2010) A new method for the analysis of germination and emergence data of weed species. Weed Research, 50, 187–198.

Reviewer #3: The manuscript is very well written and brings an important contribution to our understanding of the effect of dung beetle activity of plant recruitment. All sections of the manuscript are very clear and easy to read. I have no major concerns and only did some minor comments here below. Congratulations to the authors. That was a real pleasure to read this manuscript.

L63: Please change “de” into “the” before “probability”

Experiments 1 and 2: Could you provide some information about the fruiting period of the two plant species of the experiments? Were both species present long fruiting period since you could set up experiment 2 three and five months later experiment 1 or did you collect and then freeze the seeds?

In addition, could you add some information about rainfall during experiment 1 and 2? Did you expect variation in the dung beetle activity between these two experiments because of the different time of the year?

L201-202: The proportion of seedling establishment is a continuous variable, right? Why is a binomial structure used in this case?

L561: Legend of Figure S1: “…in 50-cm-diam plots…”

L582: “…50g of feces and beetle…”

L302: “…the results were similar…” (not “where”)

L303: Maybe you can add that you obtained the same pattern of higher seedling establishment of Poulsenia with no feces.

L384: I think it would also be relevant to add that future studies should try to set experiments enabling the disentanglement of the effect of seed burial and clustering (e.g: experiments comparing the seedling establishment of seeds at a same burial depth but at different spatial clustering).

L394-397: This sentence is not very clear. In the second part of the sentence “…possibly due to seed burial”, it is expected you give an explanation on why Poulsenia seeds present negative effects when embedded in dung, but the sentence is confused and you repeated that it is because they are embedded in dung.

Laurence Culot

6. PLOS authors have the option to publish the peer review history of their article (what does this mean?). If published, this will include your full peer review and any attached files.

Reviewer #1: No

Reviewer #2: Yes: Tanja Milotic

Reviewer #3: Yes: Laurence Culot

---

## [Author Response · Author response to Decision Letter 0]

17 Sep 2019

Dear Dr. Guillermo C. Amico,

Thank you for taking into consideration our manuscript entitled “Horizontal seed dispersal by dung beetles reduced seed and seedling clumping, but did not increase short-term seedling establishment”. After carefully considering all the comments made by the three reviewers, we are submitting our revised manuscript. Our responses (IN CAPITAL LETTERS) to all comments are detailed below. We thank the referees for their constructive suggestions. We believe that the manuscript is considerably improved and look forward to receiving your decision.

Sincerely yours,

L.A. Urrea-Galeano and E. Andresen, on behalf of all co-authors

Reviewer #1: In their manuscript “Horizontal seed dispersal by dung beetles reduced seed and seedling clumping, but did not increase short-term seedling establishment” the authors have tested the effects of horizontal seed movement by dung beetle for two tree species. They found that dung beetle activity reduced the spatial clumping of seeds and seedlings, however, it did not increase the probability of seedling establishment. This study to help fill our knowledge gaps regarding the effects of secondary seed dispersal by dung beetles. The sample design seems adequate and robust to answer their main questions. I also like writing and organization of the manuscripts as well as concise results and discussion. I present below a list of minor comments that will hopefully aid you in improving your manuscript. I therefore really welcome this ms.

THANK YOU! 

L.30- In your abstract and introduction, I think it is necessary that you state more clearly your objectives, rather than “the main goal of study was to help fill this information gap “

HE HAVE WRITTEN MORE CLEARLY THE OBJECTIVE OF OUR STUDY IN THE ABSTRACT AND INTRODUCTION SECTIONS. NOW THE ABSTRACT SAYS: “THE OBJECTIVE OF OUR STUDY WAS TO ASSESS THE EFFECTS OF DUNG BEETLE ACTIVITY ON THE SPATIAL DISTRIBUTION OF SEEDS AND SEEDLINGS, AND ON THE PROBABILITY OF SEEDLING ESTABLISHMENT.” (LINES 30-32). THE INTRODUCTION SAYS: “OUR MAIN OBJECTIVE WAS TO ASSESS, FOR TWO TREE SPECIES, THE EFFECTS OF SECONDARY SEED DISPERSAL BY DUNG BEETLES, WITH EMPHASIS ON THE HORIZONTAL MOVEMENT OF SEEDS, ON THE SPATIAL DISTRIBUTION OF SEEDS AND SEEDLINGS, AND ON THE PROBABILITY OF SEEDLING ESTABLISHMENT.” (LINES 96-98).

L. 31- I recommend adding a sentence here, explained better your field experiments

WE HAVE ADDED SOME TEXT ABOUT THE EXPERIMENTS (LINES 32-35).

L. 33 - Be specific: What tree species?

WE HAVE ADDED THE NAME OF BOTH TREE SPECIES (LINE 34).

L. 80- I think it is necessary that you state more clearly what the unique contribution of your study is. What is your study diferent from Lawson et al. (2012)? WE HAVE ADDED SOME TEXT IN THE INTRODUCTION AND IN THE DISCUSSION POINTING OUT SOME DIFFERENCES BETWEEN OUR STUDY AND LAWSON’S. PLEASE SEE LINES 87-95 (INTRODUCTON) AND LINES 369-377 (DISCUSSION). 

L. 97- Add elevation of Station DONE (LINE 110).

L. 101 - Was the experiment carried out during which season? FOR MORE CLARITY WE HAVE ADDED SOME TEXT ABOUT THE MONTHS IN WHICH THE RAINY SEASON OCCURS (LINES 114-115). WE ALSO MENTION NOW DURING WICH SEASON BOTH EXPERIMENTS WERE CARRIED OUT. FOR EXPERIMENT 1 SEE LINES 195-196 AND FOR EXPERIMENT 2 SEE LINES 202-203.

L. 107 – 108- References for frugivores are need here, or declare if it is personal observation. THANKS FOR YOUR OBSERVATION. WE HAVE ADDED THE REFERENCE (LINE 123).

L. 116 – Why did not you use a mixed of human and pig dung? FOR SEVERAL REASONS. FIRST, FOR PERSONAL REASONS, WE DO NOT USE HUMAN DUNG. SECOND, PIG DUNG HAS BEEN FOUND TO BE VERY EFFECTIVE IN ATTRACTING DUNG BEETLES (MARSH ET AL. 2013) AND IT WAS EASY TO OBTAIN THE LARGE QUANTITIES WE NEEDED FOR EXPERIMENTS. SINCE EXPERIMENTS WERE CARRIED OUT WITH PIG DUNG, WE DECIDED TO USE THE SAME BAIT FOR SAMPLING THE DUNG BEETLE COMMUNITY. WE HAVE ADDED THIS INFORMATION IN THE TEXT THAT CAN BE FOUND IN THE SUMPLEMENTARY INFORMATION; WE HAVE REMOVED THIS INFORMATION FROM THE MAIN MANUSCRIPT FOLLOWING ONE OF YOUR SUGGESTIONS.

L. 188 – Were the dung beetles identified using any entomological key? ALFONSO DÍAZ, ONE OF THE CO-AUTHORS, WHO IS EXPERT IDENTIFYING DUNG BEETLES DID THE IDENTIFICATION. IN THE SUPLEMENTARY INFORMATION WE HAVE THE FOLLOWING INFORMATION ABOUT IT: “All individuals were counted and identified in the Laboratorio de Ecoetología at the Instituto de Ecología A.C., Xalapa,Veracruz, Mexico. Also, we used a dung beetle collection with specimens collected from LTBS and Los Tuxtlas region.”

L. 121 - I would like to know more information about two focal tree species. Are these species dominant in the station? Why do you call these focal species? WE HAVED ADDED MORE INFORMATION FOR BOTH PLANT SPECIES IN THE METHODS SECTION (LINES 128-129; 132-139). WE CALL THEM FOCAL SPECIES SIMPLY BECAUSE OUR STUDY FOCUSED ON THEM; IT IS VERY COMMON TO REFER TO THE STUDY SPECIES AS THE FOCAL SPECIES.

L. 106 and L 125: Alouatta palliata or Alouatta palliata mexicana? Try to use the same name. THANKS FOR YOUR OBSERVATION. TO AVOID CONFUSION WE NOW USE THE NAME “ALOUATTA PALLIATA”, AS USED BY ESTRADA & COATES-ESTRADA (1991). (LINES 121; 141).

L.125 - Is Alouatta palliata Mexicana herbivorous or omnivore? ALOUATTA PALLIATA IS HERBIVOROUS-FRUGIVOROUS; WE HAVE ADDED SOME INFORMATION ON THIS IN LINE 120. WE HAVE ALSO ADDED THIS SAME INFORMATION FOR THE OTHER SEED-DISPERSING MAMMALS THAT CAN BE FOUND AT OUR STUDY SITE (LINE 121).

L. 131 - Why as model herbivore? Is the pig dung used as omnivores? WE HAVE REMOVED THIS TEXT AND HAVE SIMPLIFIED OUR EXPLANATION OF WHY WE USED PIG DUNG; PLEASE SEE LINES 158-160. 

L. 131 - Why you used fresh domestic pig dung rather than of Alouatta palliata? Pigs do not form part of the local biota and are onmnivores (?) rather than frugivores. IDEALLY WE WOULD HAVE USED THE DUNG OF A FOREST ANIMAL. HOWEVER, THE LOGISTICS OF FINDING ENOUGH DUNG FOR THE EXPERIMENTS MADE THIS AN UNREALISTIC PROPOSITION. UNLIKE OTHER SPECIES OF ALOUATTA, A. PALLIATA AT LOS TUXTLAS DEFECATES FROM UP IN THE CANOPY AND MUCH OF THE DUNG NEVER REACHES THE GROUND; THE SMALL AMOUNTS THAT DO, ARE DIFFICULT TO COLLECT. COMPARED TO OTHER STUDIES THAT USE COW OR HORSE DUNG (E.G. LAWSON ET AL. 2012 USED HORSE DUNG), WE THINK THAT PIG DUNG IS PROBABLY MORE SIMILAR TO DUNG THAT CAN BE FOUND IN THE FOREST.

L. 137 - I would like to see a photo of experiment in supplemental material. PHOTOS HAVE BEEN ADDED; PLEASE SEE THE SUPPLEMENTARY INFORMATION.

L. 138 – Why keep the leaf litter? TO AVOID INTERFERING WITH THE DUNG-RELOCATION BEHAVIOR OF ROLLER DUNG BEETLES. WE HAVE ADDED THE FOLLOWING EXPLANATION IN METHODS: “…we kept the leaf litter to avoid affecting the behavior of roller dung beetles, which often choose a spot hidden under litter to build their tunnel (E. Andresen, pers. obs.), and thus litter removal may cause them to roll the dung balls larger distances.” LINES 166-169.

L.136- Were the plots of all treatments covered? IN EXPERIMENT 1 WE ONLY COVERED THE PLOTS WHERE DUNG BEETLES WERE EXCLUDED. PLOTS WITH DUNG BEETLE ACCESS WERE LEFT OPEN. WE HAVE ADDED SOME TEXT TO CLARIFY (LINES 182-184).

L. 161- I think that you said the same information in the lines 163 and 164. THANKS FOR YOUR OBSERVATION. WE HAVE DELETED THE REPEATED TEXT (LINES 200-201).

L. 165 - Is the dung beetle activity went beyond the plot? Because you said that the plot with dung beetle activity was covered with netting after 48 hours. DUNG BEETLES COULD NOT RELOCATE DUNG/SEEDS OUTSIDE THE PLOT. ALL THE PLOTS WERE DELIMITED BY A FENCE. AFTER 48 H ALL DUNG HAD DISAPPEARED DUE TO DUNG BEETLE ACTIVITY, AND PLOTS WERE COVERED WITH NETTING TO AVOID SEED RAIN AND TO PREVENT LOSS OF SEEDS AND SEEDLINGS THROUGH THE ACTION OF SEED PREDATORS AND HERBIVORES. WE HAVE ADDED THE FOLLOWING TEXT AT THE END OF THE DESCRIPTION OF EXPERIMENT 1 (LINES 193-195): “Dung beetle movement was limited by the plot’s fence, i.e., seeds could not be dispersed beyond the fence. While this allowed us to find most seeds, it makes our estimates of horizontal dispersal conservative (see Discussion).” AND AGAIN, WE MENTION THIS POINT IN THE DISCUSSION (LINES 402-405). 

L. 215 - What is False Discovery Rate method? IT IS A METHOD FOR ADJUSTING P VALUES FOR MULTIPLE COMPARISONS, FOCUSING ON CONTROLLING THE FALSE DISCOVERY RATE (PROPORTION OF ALL REJECTIONS THAT ARE FALSE POSITIVES), RATHER THAN THE NUMBER OF FALSE POSITIVES PER SE, AS BONFERRONI CORRECTION DOES (JAFARI & ANSARI-POUR 2019). THIS CORRECTION CONSTITUTES AN ADEQUATE COMPROMISE BETWEEN CONTROLLING FOR FALSE POSITIVES AND FOR FALSE NEGATIVES. WE HAVE ADDED A SHORT TEXT ABOUT THIS IN LINES 259-260.

L. 218 - The reason for the dung beetle sampling should be in the methodology. THANKS FOR YOUR OBSERVATION. WE HAVE DELETED THIS INFORMATION FROM RESULTS AND ADDED IT TO SUPPLEMENTARY MATERIAL BASED ON YOUR OBSERVATION BELOW.

L 219 - 225 – In the results session should have the results regarding their objectives. I suggest you transfer this information on to supplementary material or add a question in your objectives. WE SAMPLED DUNG BEETLE AS COMPLEMENTARY INFORMATION, I.E. TO HELP US DISCUSS THE RESULTS OBTAINED FROM OUR MAIN EXPERIMENTS. SO, WE DECIDED TO TRANSFER THIS INFORMATION TO SUPPLEMENTARY MATERIAL.

L 225 - Why did you put information on Scarabaeidae, Trogidae and Silphidae? In your introduction you state that your study is about dung beetles (scarabaeinae). OUR LIST ONLY INCLUDES SCARABAEINAE. WE DO CITE A REFERENCE THAT INCLUDES THESE OTHER FAMILIES, AS WE USED IT TO OBTAIN SOME OF THE ECOLOGICAL DATA PRESENTED IN THE TABLE. IT IS TRUE THAT WE HAD LISTED ONE SPECIES (TEAMSCARABORUM OCAMPO) WHICH USED TO BE INCLUDED IN THE SCARABAEINA BUT IS NOW IN ANOHTER FAMILY AND SUBFAMILY (HYBOSORIDAE, HYBOSORINAE). SO, IN THE REVISED VERSION OF OUR TABLE WE HAVE REMOVED THIS SPECIES. SEE SUPPLEMENTARY MATERIAL.

L 233 – The sum of Pousenia was 99.9%. Check the values, please. THANKS FOR THE OBSERVATION. HE HAVE RE-CHECKED AND MADE THE CORRECTION (LINE 269).

L. 256, 258, 259 - Put the df. DONE (LINES 289-294). WE ALSO ADDED DF IN THE RESULTS OF EXPERIMENT 2 (LINES 318-323).

L. 298 - The information “but not in accordance with our prediction” should not be in the results. WE HAVE DELETED THIS SENTENCE (LINE 331).

L. 316 - In your introduction you describe the tunnelers: they construct tunnels in the soil beneath the dung pad for burying portions of dung. However, approximately 90% of dung beetle community of your study is tunneler beetles. How do you explain the large amount of seeds moved horizontally (97%)? WE HAVE RE-WRITTEN THIS PART IN THE INTRODUCTION, TO DESCRIBE BETTER THE DUNG-RELOCATION BEHAVIOR OF TUNNELERS AND ROLLERS; PLEASE SEE LINES 55-59. ALSO, IN THE DISCUSSION WE HAVE ADDED THE FOLLOWING TEXT: “On the other hand, tunneler dung beetles bury the dung relatively close to the dung source [10], but that does not mean that they do not cause any horizontal movement of seeds. Tunnelers sometimes push dung portions on the soil surface for a short distance (<20 cm) before burying it; furthermore, underground tunnels are often dug with an angle <90°, such that seeds moved into the tunnels are displaced both vertically and horizontally (L.A. Urrea-Galeano and E. Andresen, pers. obs.).” LINES 407-412.

L. 325 - I would like you to discuss a bit about what time would be enough. Will this depend on the plant species? WE AGREE THAT THIS TIME WILL DEPEND ON THE PLANT SPECIES. WE HAVE REWRITTEN THE END OF OUR FIRST PARAGRAPH OF THE DISCUSSION AND DO NOT MENTION HERE THIS ISSUE ANYMORE (PLEASE SEE LINES 355-357); HOWEVER WE MENTION IT NOW IN A DIFFERENT PART, AND HAVE ADDED THIS TEXT: “Furthermore, since the life-stage at which plants suffer negative density-dependent mortality varies among species and contexts, future research will need to carefully determine the necessary duration of studies, in order to accurately assess the effects of reduced seed/seedling clumping caused by dung beetles, on plant fitness.” SEE LINES 424-428. 

Reviewer #2: Review report PONE-D-19-15149

General comments

The paper covers an interesting topic and certainly fills a gap in knowledge as it is indeed assumed that dung beetles have a positive effect on seed germination and seedling survival. I am not too familiar with the most recent publications in tropical dung beetle ecology, but the paper certainly fills a gap in the current knowledge. THANK YOU!

The manuscript is well written in a clear and concise matter and the figures are clear. The used methods are correct, although I do have a suggestion for improving the statistical analysis. I am not sure whether the method is applicable on the dataset that was collected (no raw data were provided), but I’m quite certain that a survival analysis would give more insight in the establishment experiment (experiment 2). Now the authors analyzed the peak emergence data for Bursera, because a large portion of the seedlings died at a given moment. In a survival analysis, the cumulative probability of a certain event (establishment in this case) is calculated. Onofri et al. (2010) wrote a very interesting paper about using survival analysis for the analysis of the germination and emergence of seeds. There is also an R-package available on CRAN with functions for survival analyses (package ‘survival’). By conducting a survival analysis, the authors could assess the differences between treatments over the entire time series instead of using 1 single moment in a long time series. THANK YOU FOR THIS SUGGESTION. WE HAVE READ THE PAPER FROM ONOFRI ET AL. (2010) AND SOME OTHER LITERATURE ABOUT THIS ISSUE, AND WE DECIDED TO FOLLOW YOUR SUGGESTION AND TO RE-ANALYSE OUR DATA FOR SEEDLING ESTABLISHMENT (EXPERIMENT 2) USING A SURVIVAL ANALYSIS THAT CONSIDERS RANDOM EFFECTS (FOLLOWING AUSTIN 2017). THUS, WE HAVE ELIMINATED FROM THE MANUSCRIPT ALL THE INFORMATION OF OUR PRIOR ANALYSES DONE WITH GLMMS. NOW, YOU CAN FIND IN THE ANALYSIS AND RESULTS SECTIONS JUST THE INFORMATION ABOUT THE SURVIVAL ANALYSIS (LINES 244-250 AND LINES 328-335). WE HAVE ALSO CHANGED FIGURE 2, WHICH NOW CONTAINS THE SURVIVAL CURVES (LINES 337; 343-344). IT IS IMPORTANT TO NOTE THAT OUR RESULTS FROM THE SEEDLING ESTABLISHMENT WITH THE SURVIVAL ANALYSIS ARE CONSISTENT WITH THOSE OBTAINED WITH THE GLMMS. THEREFORE, OUR GENERAL CONCLUSIONS ABOUT THE EFFECT OF DUNG AND BEETLE ACTIVITY ON THIS RESPONSE VARIABLE ARE THE SAME AS BEFORE. 

Data files are provided for all experiments and metadata are provided. However, the raw datasets are not provided, instead derived data (indices, cumulated dung beetle numbers without capture dates) are given. I don’t know the journal’s policy about data availability and the motivation for providing data, but these types of datasets are not very useful. Maybe, the results of this study could become useful in a future study doing a meta-analysis which needs raw data (e.g., real distances instead of the Clark Evans index). If the authors cannot be contacted at that time or they cannot retrieve the raw datasets anymore, sharing the datasets with this paper was not worth the effort. YOU ARE RIGHT. WE HAVE PROVIDED THE RAW DATA NOW IN SUPPORTING INFORMATION.

Also, the dung beetle dataset could provide interesting information for policy makers and conservationalists (e.g., for making red lists), so it would be useful to put it in a suitable database. For biodiversity data, the ‘Global biodiversity information facility (GBIF)’ (www.gbif.org) is the standard database. When publishing data at GBIF, you become the author of the dataset and you get a doi code. So, the dataset can be cited and you get acknowledged for your efforts. YOU ARE RIGHT. WE HAVE PROVIDED THE RAW DATA NOW IN SUPPORTING INFORMATION.

Specific comments

L43: potential advantage to plants of the second phase of… -> remove ‘to plants’ as it is clear that plants are the beneficiary of the process DONE (LINE 45)

L61: ‘sign’ sounds a bit weird -> maybe ‘nature’ or ‘(final) result’ ? INSTEAD OF “SIGN”, WE WROTE “DIRECTION” (LINE 65)

L117: Which measures were taken? Which equipment was used and what was the precision of the measurements? THANKS FOR THE OBSERVATION. ACTUALLY, WE DID NOT MEASURE DUNG BEETLES. IT WAS A MISTAKE TO WRITE THIS. WE GOT MEASUREMENTS FROM DÍAZ AND FAVILA 2006. WE HAVE RE-WRITTEN THIS SENTENCE. SEE TEXT IN SUPPLEMENTARY INFORMATION. THIS PART IS NOW IN SUPPLEMENTARY INFORMATION, FOLLOWING ONE COMMENT FROM REVIEWER 1.

L139: How did you decide to use 20 seeds in dung portions of 50g? How is this related to the seed density in ‘natural’ droppings found in this region? IN METHODS WE HAVE ADDED THE FOLLOWING TEXT TO ADDRESS THIS POINT: “Since the amount of seeds present in the defecations of rainforest mammals can vary tremendously, depending on the plant and animal species (e.g., [8,38,39], we used seed numbers that can commonly be found in howler-monkey dung piles (e.g., [40,41])”. PLEASE SEE LINES 173-176.

L152: seed ‘alive’: did you test the viability of the seeds? If not, intact seeds would be a better description. WE DID NOT TEST THE VIABILITY OF THE SEEDS. WE HAVE CHANGED THE WORD “ALIVE” FOR “INTACT” THROUGHOUT THE MANUSCRIPT (LINES 189, 190).

L159 and following: It is not clear from the text, but did you count the number of Poulsenia and Bursera seedlings originating from the soil seed bank? TO ADDRESS THIS POINT WE HAVE ADDED THE FOLLOWING TEXT IN THE METHODS SECTION: “We assumed that all seedlings of the focal plant species that we recorded, originated from our experimental seeds because: (i) all plots were > 10 m away from any fruiting adult, and (ii) in a previous study in the same sites and with the same treatments, only two seedlings of Bursera and two of Poulsenia established, overall, from the soil seed bank during a time period of 8 months [33].” SEE LINES 211-216. 

L229-232: this part rather belongs to the discussion section. THANK YOU FOR THE OBSERVATION. WE HAVE RE-WRITTEN THIS PARAGRAPH. LINES 266-267.

L232: See my earlier remark. Unless you have tested the retrieved seeds for viability, you should refer to them as intact seeds. DONE (LINE 268). 

L244-246: this rather belongs to the discussion section. WE MOVED THIS INFORMATION TO THE DISCUSSION SECTION (LINES 402-405).

L319-323: It seems that the presence of dung did have a greater effect on the emergence of seedlings than the presence of dung beetles. The fact that dung reduces the germinability of seeds was also found in other studies, e.g., Milotić and Hoffmann (2016). This study was done in a completely different environment and using different plant species (temperate grassland species), but I am not aware of similar studies in a tropical environment. THANKS FOR THE OBSERVATION. WHAT YOU SAY IS RIGHT AND WE NOW INCLUDE SOME TEXT ON THIS TOWARDS THE END OF OUR DISCUSSION (LINES 454-457).

L338: Another factor to consider is the fact that clean, undigested seeds were used in the experiments. I assume that was practically impossible to use digested seeds as you would have to feed wild animals with seeds and collect the dung afterwards (furthermore, the issue would arise about which species should be used as a ‘digester’ species), but it is worthwhile mentioning this in the discussion. In the natural situation, seeds that passed the gut environment of a frugivore might have an altered germination probability and speed. So, maybe the presence of dung would not be that much of a problem then. THANKS FOR YOUR OBSERVATION. WE SEE THIS MORE AS A METHODOLOGICAL LIMITATION; SO, RATHER THAN INCLUDING IT IN THE DISCUSSION (WHICH HAS ICREASED IN LENGTH IN THE REVISED MANUSCRIPT), WE MENTION THIS ISSUE IN THE METHODS, AT THE END OF THE DESCRIPTION OF EXPERIMENT 2, WHERE WE HAVE ADDED THIS TEXT: “Finally, we acknowledge that using seeds extracted from fruits may yield different results compared to using seeds that have passed through the digestive system of a mammal. However, we expect that whatever difference there might be in terms of seed germination would equally have affected our three treatment levels.” SEE LINES 216-219. 

L350 and following: did you test the viability of the seeds prior to the experiments? It could also be that a relatively large proportion of the seeds is dead or dormant. Is there anything known about the germination ecology of these species (e.g., in Baskin and Baskin (2001))? AS MENTIONED ABOVE, WE DID NOT TEST THE VIABILITY OF SEEDS. FROM LITERATURE WE FOUND THAT SEEDS OF THESE TWO SPECIES ARE NOT DORMANT; WE HAVE ADDED THIS INFORMATION IN METHODS (LINES 141-142). IF THE SPECIES HAVE NATURALLY LOW SEED VIABILITY, IT WOULD HAVE AFFECTED ALL OUR TREATMENTS SIMILARLY, AND THUS THE PATTERNS OBSERVED, IN TERMS OF THE EFFECTS OF DUNG BEETLES, WOULD HAVE BEEN THE SAME.

L393: What is the usual fruiting season for these species or do they carry fruits year-round? FRUITING SEASON FOR BURSERA IS FROM OCTOBER TO MAY AND FOR POULSENIA IS FROM MAY TO NOVEMBER. WE HAVE ADDED THIS INFORMATION IN THE METHODS SECTION (LINES 133; 137).

Fig S3: For Bursera, the mean nearest neighbor distance in the +feces+beetles treatment increases over time. Do you have any idea why? BURSERA SEEDLINGS DIED SHORTLY AFTER ESTABLISHING, DUE TO SHADE-INTOLERANCE; WE BELIEVE THAT THE INCREASE IN NEIGHBOR DISTANCE WAS SIMPLY A RESULT OF THERE BEING FEWER SEEDLINGS. PLEASE, NOTE THIS FIGURE HAS CHANGED FROM Fig S3 TO S4 Fig.

Reviewer #3: The manuscript is very well written and brings an important contribution to our understanding of the effect of dung beetle activity of plant recruitment. All sections of the manuscript are very clear and easy to read. I have no major concerns and only did some minor comments here below. Congratulations to the authors. That was a real pleasure to read this manuscript. THANK YOU!

L63: Please change “de” into “the” before “probability” DONE (LINE 67).

Experiments 1 and 2: Could you provide some information about the fruiting period of the two plant species of the experiments? WE HAVE ADDED THIS INFORMATION IN THE METHODS SECTION (LINES 133; 137). 

Were both species present long fruiting period since you could set up experiment 2 three and five months later experiment 1 or did you collect and then freeze the seeds? FOR BOTH EXPERIMENTS WE COLLECTED ALL EXPERIMENTAL SEEDS ONCE BY THE TIME EXPERIMENT 2 STARTED (SEEDLING EXPERIMENT). SINCE WE NEEDED FRESH SEEDS FOR THIS EXPERIMENT (TO AVOID ANY INTERFERENCE IN GERMINATION), WE STARTED WITH THIS EXPERIMENT FIRST USING HALF OF THE SEEDS COLLECTED. THE REST OF SEEDS WAS DRIED AND STORED FOR EXPERIMENT 1 (SEEDS EXPERIMENT). ALTHOUGH FOR EXPERIMENT 1 WE COULD HAVE OBTAINED FRESH SEEDS DIRECTLY FROM TREES, WE DECIDED TO USE THE SEEDS ALREADY STORED TO FACILITATE THE MARKING PROCESS OF SEEDS REQUIRED FOR THIS EXPERIMENT. WE HAVE ADDED SOME TEXT IN THE METHODS SECTION TOO (LINES 146-155).

In addition, could you add some information about rainfall during experiment 1 and 2? WE HAVED ADDED INFORMATION ABOUT THE RAINFALL PATTERNS (LINES 113-115), AND WE ALSO MENTION THE SEASON DURING WHICH EACH EXPERIMENT WAS DONE (LINES 195-196 AND 202-203). 

Did you expect variation in the dung beetle activity between these two experiments because of the different time of the year? BASED ON OUR DUNG BEETLE SAMPLING (SEE ALSO ESTRADA & COATES-ESTRADA, 1991) WE KNOW THAT THERE COULD BE A VARIATION IN THE MEAN NUMBER OF SPECIES AND INDIVIDUALS CAPTURED PER TRAP, WHICH IS RELATED WITH THE TIME OF THE YEAR. THUS, WE COULD EXPECT A VARIATION OF THE DUNG BEETLE ACTIVITY BETWEEN THESE TWO EXPERIMENTS. HOWEVER, DUE TO THE LOW PERCENTAGE OF ROLLERS CAPTURED BY THE TIME EXPERIMENT 2 AND EXPERIMENT 1 STARTED, WE DO NOT BELIEVE THAT THIS VARIATION COULD AFFECT OUR GENERAL CONCLUSION REGARDING THEIR EFFECT ON THE SPATIAL DISTRIBUTION OF SEED AND SEEDLINGS. 

L201-202: The proportion of seedling establishment is a continuous variable, right? Why is a binomial structure used in this case? SEEDLING ESTABLISHMENT CAN BE ANALYZED USING A BINOMIAL DISTRIBUTION IN WHICH THE NUMBER OF TRIALS IS THE INITIAL NUMBER OF SEEDS AND THE SUCCESSES ARE THE NUMBER OF ESTABLISHED SEEDLINGS. THE PARAMETER ESTIMATED FROM SUCH MODEL IS P, THE PROPORTION OF SUCCESSES (PROPORTION OF SEEDLINGS ESTABLISHED IN THIS CASE). HOWEVER, FOLLOWING A SUGGESTION BY REVIEWER 2 WE HAVE RE-ANALYSED THESE DATA WITH A SURVIVAL ANALYSIS TO INCLUDE THE DATA FROM OVER THE ENTIRE EXPERIMENT, NOT JUST THE PEAK OF ESTABLISHMENT. 

L561: Legend of Figure S1: “…in 50-cm-diam plots…” DONE (LINE 683). PLEASE NOTE THAT Figure S1 IS NOW S2 FIG.

L582: “…50g of feces and beetle…” THANKS FOR THE OBSERVATION. HOWEVER, WE HAVE DELETED THIS FIGURE FROM THE MANUSCRIPT AND REPLACED IT WITH A FIGURE OF SURVIVAL CURVES, ACCORDING TO THE WAY IN WHICH WE RE-ANALYSED THESE DATA.

L302: “…the results were similar…” (not “where”) WE HAVE RE-WRITTEN THIS PARAGRAPH (LINES 328-335) BASED ON A COMMENT FROM REVIEWER 2.

L303: Maybe you can add that you obtained the same pattern of higher seedling establishment of Poulsenia with no feces. THANK YOU FOR THIS OBSERVATION. HOWEVER, SINCE WE RE-ANALYSED THESE DATA USING A SURVIVAL ANALYSIS, WE HAVE DELETED THIS INFORMATION FROM THE TEXT. 

L384: I think it would also be relevant to add that future studies should try to set experiments enabling the disentanglement of the effect of seed burial and clustering (e.g: experiments comparing the seedling establishment of seeds at a same burial depth but at different spatial clustering). DONE (LINES 441-442)

L394-397: This sentence is not very clear. In the second part of the sentence “…possibly due to seed burial”, it is expected you give an explanation on why Poulsenia seeds present negative effects when embedded in dung, but the sentence is confused and you repeated that it is because they are embedded in dung. THANKS FOR YOUR OBSERVATION. WE HAVE RE-WRITTEN THIS SENTENCE FOR CLARITY (LINES 452-454).

---

## [Decision Letter · Decision Letter 1]

4 Oct 2019

PONE-D-19-15149R1

Horizontal seed dispersal by dung beetles reduced seed and seedling clumping, but did not increase short-term seedling establishment

PLOS ONE

Dear Dra. Lina Adonay Urrea Galeano,

Thank you for submitting your manuscript to PLOS ONE. After careful consideration, we feel that it has merit but does not fully meet PLOS ONE’s publication criteria as it currently stands. Therefore, we invite you to submit a revised version of the manuscript that addresses the points raised during the review process.

The reviewers coincide that comments have been addressed and manuscript has improved considerably. However, there are some minors concerns (see Reviewer #3) the authors should address. 

We would appreciate receiving your revised manuscript by October 18. To enhance the reproducibility of your results, we recommend that if applicable you deposit your laboratory protocols in protocols.io, where a protocol can be assigned its own identifier (DOI) such that it can be cited independently in the future. For instructions see: http://journals.plos.org/plosone/s/submission-guidelines#loc-laboratory-protocols

We look forward to receiving your revised manuscript.

Kind regards,

Guillermo C. Amico

Academic Editor

PLOS ONE

Additional Editor Comments (if provided):

The reviewers coincide that comments have been addressed and manuscript has improved considerably. However, there are some minors concerns that the authors should address.

Reviewers' comments:

Reviewer's Responses to Questions

**Comments to the Author**

1. If the authors have adequately addressed your comments raised in a previous round of review and you feel that this manuscript is now acceptable for publication, you may indicate that here to bypass the “Comments to the Author” section, enter your conflict of interest statement in the “Confidential to Editor” section, and submit your "Accept" recommendation.

Reviewer #1: All comments have been addressed

Reviewer #2: All comments have been addressed

Reviewer #3: All comments have been addressed

2. Is the manuscript technically sound, and do the data support the conclusions?

Reviewer #1: Yes

Reviewer #2: Yes

Reviewer #3: Yes

3. Has the statistical analysis been performed appropriately and rigorously? 

Reviewer #1: Yes

Reviewer #2: Yes

Reviewer #3: Yes

4. Have the authors made all data underlying the findings in their manuscript fully available?

Reviewer #1: Yes

Reviewer #2: Yes

Reviewer #3: Yes

5. Is the manuscript presented in an intelligible fashion and written in standard English?

Reviewer #1: Yes

Reviewer #2: Yes

Reviewer #3: Yes

6. Review Comments to the Author

Reviewer #1: (No Response)

Reviewer #2: This revised version of the manuscript has improved considerably. The methods are written down in a way it is possible to reproduce the experiment, and I really appreciate the pictures that are included as an appendix. The whole text is well written, and the images are clear. All suggestions and corrections I proposed were addressed properly, and questions were answered. After carefully reading the manuscript, inspecting the graphs and appendices, no new issues raised, so I would advise to publish this paper.

Reviewer #3: You responded satisfactorily to most of my comments and I think that the manuscript has been well improved.

I still think that you can expect a different secondary seed dispersal activity between seasons, and therefore between the experiment 1 and 2. You may have a low proportion of rollers, but the tunnelers also influence the burial depth, which consequently can affect the probability of seedling emergence. This difference does not discredit your work, but I think that a mention of this possible bias should be mentioned in the discussion.

Please find a few minor comments based on the reading of this new version of the manuscript:

L197-198: What do you mean by “conservative” in this case. The restricted area of your experiments can bias the results towards an underestimation of the horizontal distances. Is it what you mean? If so, I suggest to rephrase. If not, please clarify or delete the sentence and keep this information for the discussion.

L205: I suggest writing “one of the driest month”

L2016: I suggest writing “the beginning of the rainy season”

7. PLOS authors have the option to publish the peer review history of their article (what does this mean?). If published, this will include your full peer review and any attached files.

Reviewer #1: No

Reviewer #2: Yes: Tanja Milotic

Reviewer #3: Yes: Laurence Culot

---

## [Author Response · Author response to Decision Letter 1]

9 Oct 2019

Dear Dr. Guillermo C. Amico,

We have read and addressed all the comments made by Dr. Laurence Culot (Reviewer #3); please see our responses (in CAPITAL LETTERS) to the comments detailed below. Once again, we thank you and the referees for all comments done to improve our manuscript.

Sincerely yours,

L.A. Urrea-Galeano and E. Andresen, on behalf of all co-authors

Reviewer #1: (No Response)

Reviewer #2: This revised version of the manuscript has improved considerably. The methods are written down in a way it is possible to reproduce the experiment, and I really appreciate the pictures that are included as an appendix. The whole text is well written, and the images are clear. All suggestions and corrections I proposed were addressed properly, and questions were answered. After carefully reading the manuscript, inspecting the graphs and appendices, no new issues raised, so I would advise to publish this paper. THANK YOU!

Reviewer #3: You responded satisfactorily to most of my comments and I think that the manuscript has been well improved. THANK YOU!

I still think that you can expect a different secondary seed dispersal activity between seasons, and therefore between the experiment 1 and 2. You may have a low proportion of rollers, but the tunnelers also influence the burial depth, which consequently can affect the probability of seedling emergence. This difference does not discredit your work, but I think that a mention of this possible bias should be mentioned in the discussion. THANKS FOR YOUR OBSERVATION, WE AGREE. WE HAVE MENTIONED THIS ISSUE IN TWO PARTS IN THE DISCUSSION. IN LINES 401-406 WE HAVE WRITTEN THIS: “However, it could also be that seasonal differences in dung beetle assemblages caused a higher proportion of seeds to be buried and/or buried more deeply in Experiment 2, which was setup in April-June, than in Experiment 1, which was done in October. When we sampled dung beetles, we found higher abundances, particularly of large beetles (≥ 10 cm body length), in April than in October (S1 Table).” ALSO, IN LINES 478-483 WE HAVE ADDED THIS TEXT: “For example, differences between species could be due to seasonality in dung beetle activity, since the experiment for Bursera was started in April and the one with Poulsenia in June, a drier and a rainier month, respectively. Secondary seed dispersal by dung beetles has been shown to be affected by seasonality [7], not only because dung beetle assemblages vary seasonally [10], but also because in rainier months softer soils might favor deeper seed burial [21], which in turn might hinder seedling emergence.” FINALLY, IN S1 TABLE WE HAVE ADDED INFORMATION ON THE NUMBER OF INDIVIDUALS CAPTURED IN EACH OF THE SAMPLINGS, SO THAT THE POTENTIAL EFFECT OF SEASONALITY CAN BE BETTER APPRECIATED.

Please find a few minor comments based on the reading of this new version of the manuscript:

L197-198: What do you mean by “conservative” in this case. The restricted area of your experiments can bias the results towards an underestimation of the horizontal distances. Is it what you mean? If so, I suggest to rephrase. If not, please clarify or delete the sentence and keep this information for the discussion. YES, UNDERESTIMATION IS WHAT WE MEAN; WE HAVE REWRITTEN THE SENTENCE FOR CLARITY (LINES 197-198). NOW IT READS: “While this allowed us to find most seeds, it probably caused some underestimation of horizontal distances (see Discussion).”

L205: I suggest writing “one of the driest months” DONE (LINE 205)

L2016: I suggest writing “the beginning of the rainy season” DONE (LINE 206)

---

## [Editor Report · Decision Letter 2]

14 Oct 2019

Horizontal seed dispersal by dung beetles reduced seed and seedling clumping, but did not increase short-term seedling establishment

PONE-D-19-15149R2

Dear Dra. Lina Adonay Urrea Galeano,

We are pleased to inform you that your manuscript has been judged scientifically suitable for publication and will be formally accepted for publication once it complies with all outstanding technical requirements.

With kind regards,

Guillermo C. Amico

Academic Editor

PLOS ONE
---

## [Editor Report · Acceptance letter]

15 Oct 2019

PONE-D-19-15149R2 

Horizontal seed dispersal by dung beetles reduced seed and seedling clumping, but did not increase short-term seedling establishment 

Dear Dr. Urrea Galeano:

I am pleased to inform you that your manuscript has been deemed suitable for publication in PLOS ONE. Congratulations! Your manuscript is now with our production department. 

With kind regards,

on behalf of

Dr. Guillermo C. Amico 

Academic Editor

PLOS ONE